# Early life inflammation is associated with spinal cord excitability and nociceptive sensitivity in human infants

Maria M. Cobo [1,2], Gabrielle Green [1], Foteini Andritsou[1], Luke Baxter [1], Ria Evans Fry[1,3], Annika Grabbe [4], Deniz Gursul[1], Amy Hoskin[1], Gabriela Schmidt Mellado[1], Marianne van der Vaart [1], Eleri Adams[1,3], Aomesh Bhatt [1], Franziska Denk [5], Caroline Hartley [1] & Rebeccah Slater [1✉]

Immune function and sensitivity to pain are closely related, but the association between early life inflammation and sensory nervous system development is poorly understood—especially in humans. Here, in term-born infants, we measure brain activity and reflex withdrawal activity (using EEG and EMG) and behavioural and physiological activity (using the PIPP-R score) to assess the impact of suspected early-onset neonatal infection on tactile- and noxious-evoked responses. We present evidence that neonatal inflammation (assessed by measuring C-reactive protein levels) is associated with increased spinal cord excitability and evoked brain activity following both tactile and noxious stimulation. There are early indications that this hyperalgesia could be maintained post-inflammation, supporting pre-clinical reports of early-life immune dysfunction influencing pain sensitivity in adults.

[1] Department of Paediatrics, University of Oxford, Oxford, UK. [2] Universidad San Francisco de Quito USFQ, Colegio de Ciencias Biologicas y Ambientales, Quito, Ecuador. [3] Newborn Care Unit, John Radcliffe Hospital, Oxford University Hospitals NHS Foundation Trust, Oxford, UK. [4] Paracelsus Medical University, Nuremberg, Germany. [5] Wolfson Centre for Age-Related Diseases, Institute of Psychiatry, Psychology & Neuroscience, King's College London, London, UK. ✉email: rebeccah.slater@paediatrics.ox.ac.uk

Our nociceptive and immune systems are intimately intertwined, working together to protect us from injury or disease[1]. In infants and children, both systems undergo extensive fine-tuning and maturation[2,3]. Adverse events during this delicate developmental process can have deleterious and long-lasting consequences. For example, deficiencies in the exposure to certain microorganisms can pre-dispose us to immune-mediated diseases[2] like asthma and type-1 diabetes[4], while repeated painful procedures in neonates may negatively affect nociceptor sensitivity[5] and have been associated with altered behavioural, motor, and cognitive neurodevelopment[6,7].

How the development of these two systems influences each other is much less clear, even though it is a highly relevant clinical question. Suspected infection is very prevalent in neonates[8], affecting 13–20% of infants in the postnatal wards[9], yet the consequences that such early-life inflammation may have on human nociceptive circuitry are not known[10]. Evidence from the preclinical literature suggests that neonatal inflammation might have very serious consequences[11]. There are, for instance, reports that experimental activation of the neonatal immune system in rat pups, via lipopolysaccharide (LPS) administration, causes long-term increased pain sensitivity[12–16]. However, studies investigating how the acute phase of early-life inflammation impacts pain sensitivity are scarce. Moreover, given the many known inter-species differences in both immune and nociceptive systems[17], it is of vital importance that we generate data in human neonates, which can be relevantly back-translated into future rodent models.

Here, we have generated such human neonatal data—capitalising on our team's access to a rare resource: well-matched cohorts of newborn infants (with similar perinatal histories) who can be differentiated based on the C-reactive protein (CRP) levels in their blood (Fig. 1). We studied neonates who presented with risk factors for suspected early-onset neonatal infection (see[18] for guidelines that were in place during the study period) and received prophylactic antibiotic treatment (Table 1). They were grouped according to the level of CRP in their blood or other laboratory or clinical evidence of infection within 24 h from the start of antibiotic treatment. The 'Neonatal Inflammation Group' presented with CRP > 10 mg/l or evidence of infection, whereas the 'Neonatal Control Group' presented with CRP < 10 mg/l and no evidence of infection. The threshold of 10 mg/l was selected because in our local unit prophylactic antibiotic treatment was discontinued 36-hours after the first dose if the CRP was below 10 mg/l and there were no other clinical or laboratory signs of infection (Fig. 1)[18].

We hypothesised that neonatal inflammation causes increased spinal cord excitability and increased cortical brain activity in response to (i) a noxious heel lance, which was medically required to establish the level of CRP in the blood, and (ii) in response to non-noxious tactile stimulation to the heel. We tested both these hypotheses by comparing the magnitude of spinal cord mediated reflex withdrawal activity assessed using electromyography (EMG) and stimulus-evoked changes in brain activity assessed using electroencephalography (EEG) in the Neonatal Inflammation Group and the Neonatal Control Group.

## Results and discussion

**Inflammation is associated with increased reflex withdrawal and brain activity in response to a heel lance.** Newborn infants in the Neonatal Inflammation Group had significantly greater spinal cord mediated reflex withdrawal EMG activity and noxious-evoked EEG brain activity in response to heel lancing (within 24 h from presentation of risk factors) compared to neonates who do not have raised inflammatory markers (magnitude of noxious-evoked reflex withdrawal activity: Neonatal Inflammation Group: $n = 21$, mean = 42.1, SD = 30.9; Neonatal Control Group: $n = 36$, mean=30.2, SD = 20.6; $\Delta = 11.9$, 95%CI = [1.32, ∞], $t$ test $t = 1.74$, $p = 0.048$; magnitude of noxious-evoked brain activity: Neonatal Inflammation Group: $n = 19$, mean=1.04, SD = 0.68; Neonatal Control Group: $n = 32$, mean = 0.64, SD = 0.52; $\Delta = 0.41$, 95% CI = [0.13, ∞], $t = 2.4$, $p = 0.022$, $p$ values corrected for multiple comparisons using Holm's method, Table 2, Fig. 2). Due to the continuous nature of the CRP level variable, we assessed correlations between CRP level and EMG and EEG responses. We show that the spinal cord mediated reflex withdrawal activity and noxious-evoked brain activity significantly correlated with the level of CRP in the blood sample (reflex withdrawal activity: $n = 57$, Pearson correlation $r = 0.28$, $p = 0.028$, noxious-evoked brain activity: $n = 51$, $r = 0.31$, $p = 0.023$, Fig. 2). Taken together, the combined evidence of significant correlations with CRP level and significant between-group differences (when CRP level is dichotomised at the clinical decision-making threshold of 10 mg/l), suggest these effects are clinically meaningful, despite the modest $p$ values.

Our findings are thus in line with evidence from adult humans and animals, which shows that neuro-immune interactions increase spinal cord excitability and nociceptive sensitivity[19–22], albeit substantial evidence from both human and animal neonates is lacking. Pain assessment during the period of neonatal inflammation has not been extensively investigated and studies undertaken in neonatal rat pups report conflicting results[14,22,23]. Our specific findings in human infants using CRP as a marker of inflammation could be further explored in rodent models to elucidate the potential mechanisms of these neuro-immune interactions.

In contrast, clinical pain scores (calculated using the well-validated Premature Infant Pain Profile-Revised [PIPP-R]) were not significantly altered by the presence of inflammation (PIPP-R score: Neonatal Inflammation Group: $n = 21$, mean = 5.7, SD = 3.3; Neonatal Control Group: $n = 34$, mean = 6, SD = 3.6; $t = -0.32$, $p = 0.64$). Given that an increase in reflex withdrawal and noxious-evoked brain activity were observed in the presence of inflammation, an increase in PIPP-R scores may also have been expected; however, this was not observed (Supplementary Table 1 shows the relationship between CRP levels and the PIPP-R scale individual factors). While it is possible that we are underpowered to observe an effect on clinical pain scores (given the subjective nature of PIPP-R assessments in contrast to objective EMG/EEG assessments), our observation is consistent with clinical observations in adults and neonates, as well as data from animal studies, which demonstrate that inflammation suppresses behavioural activity, causing listlessness and lethargy[24,25]. These protective inhibitory behavioural responses to inflammation, often reported being a key symptom of 'sickness syndrome'[26], likely provide a mechanism whereby available energy can be directed towards fighting pathogens rather than classic exploratory behaviours and social interactions[24].

Increased lethargy and inhibition of behavioural activity will be highly influential on the magnitude of clinical pain scores, as these scores are predominantly based on observing behaviours[27]. This potentially limits the utility of these scores in neonates with pathological conditions, where motor inhibition in response to aversive stimulation may be a critical mechanism to conserve energy. Further research is warranted to establish how the presence of neonatal inflammation impacts clinical pain scores and the degree to which subjective observational assessment of infant pain-related behaviours might influence pain scoring.

**Inflammation is associated with increased reflex withdrawal and brain activity in response to tactile stimulation.** In neonates clinical signs of 'sickness syndrome' also include increased

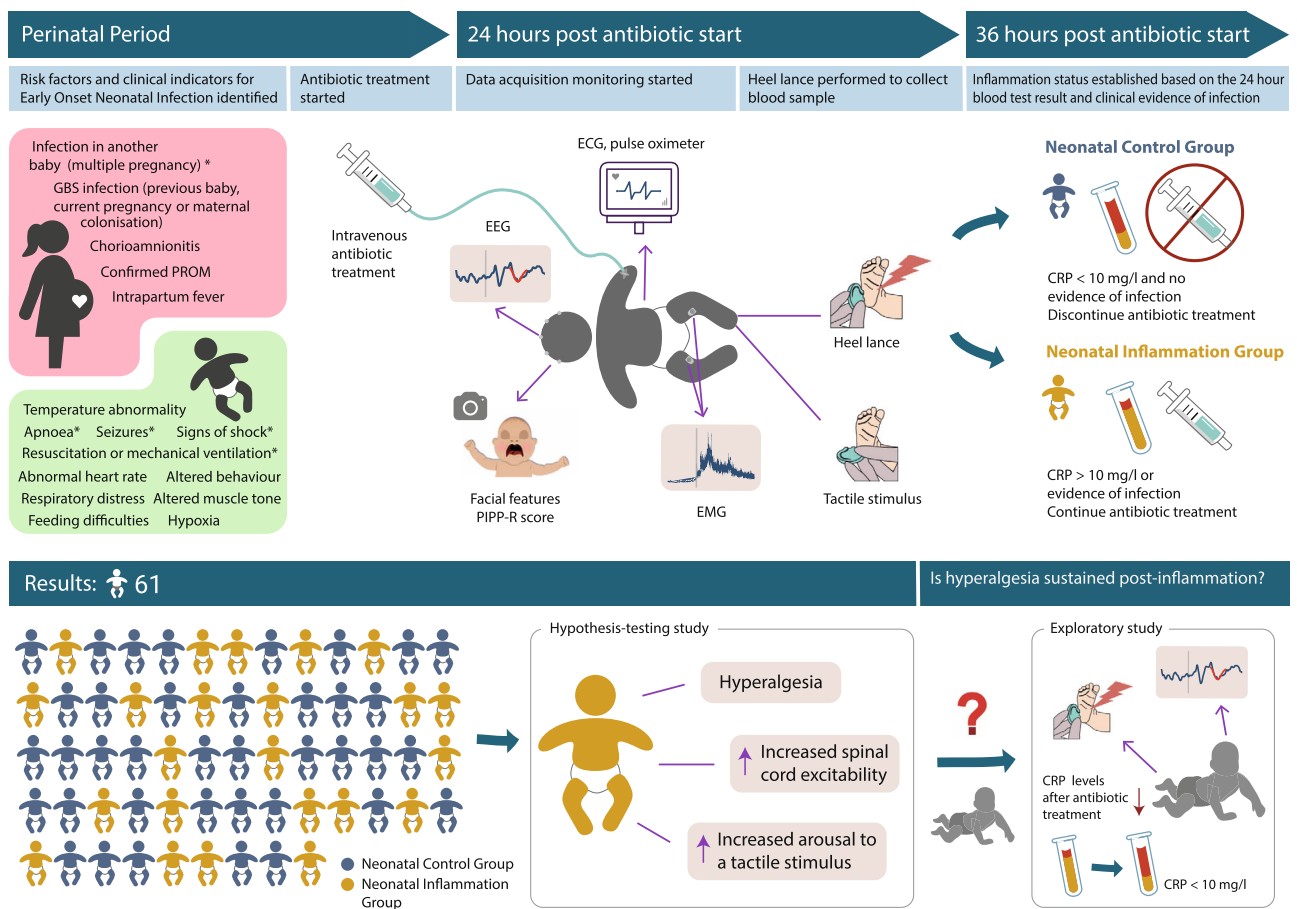

**Fig. 1 Study design and main results.** *Red flag risk factors and clinical indicators for Early-Onset Neonatal Infection (EONI). Antibiotic treatment was started if any red flag or two or more non-red-flag risk factors or clinical indicators were identified. See ref. [18] for the complete National Institute for Health and Care Excellence (NICE) EONI screening guidelines in place during the study period. *GBS* Group B *Streptococcus*, *PROM* premature rupture of membranes, *EEG* electroencephalography, *PIPP-R* premature infant pain profile-revised, *EMG* electromyography, *ECG* electrocardiogram, *CRP* C-reactive protein.

irritability and restlessness[28]. This suggests that in the absence of nociceptive input infants display more erratic body movements, which could be mediated by increased spinal cord excitability. To test the hypothesis that neonatal inflammation increases spinal cord excitability and arousal, we assessed whether the presence of neonatal inflammation (i) increased spinal cord mediated reflex withdrawal activity and (ii) increased arousal-related brain activity[29] following tactile stimulation (See Supplementary Figure 1). As hypothesised, neonatal inflammation evoked increased reflex withdrawal activity to tactile stimulation (magnitude of reflex withdrawal: Neonatal Inflammation Group: $n = 21$, mean = 17.9, SD = 12.2; Neonatal Control Group: $n = 36$, mean = 11.8, SD = 12; $\Delta = 6.15$, 95%CI = [0.92, ∞], $t = 1.85$, $p = 0.037$, Fig. 3a) and significantly greater brain activity following tactile stimulation (magnitude of tactile-evoked brain activity: Inflammation Group, $n = 19$, mean=0.1, SD = 0.09; Control Group, $n = 29$, mean=0.05, SD = 0.05; $\Delta = 0.05$, 95% CI = [0.02, ∞], $t = 2.66$, p = 0.007, Fig. 3b).

Amplification of muscle reflexes in the presence of inflammation is likely mediated by increased excitability within dorsal horn networks[30], as well as potential descending influences from central brain networks[31,32]. This evidence that cerebral arousal responses are increased in the presence of neonatal inflammation likely underpins clinical observations of restlessness in infants presenting with inflammation.

**Exploratory data suggests that noxious-evoked brain activity remains elevated post-treatment for inflammation.** The primary purpose of our hypothesis-testing study was to examine the acute consequences of neonatal inflammation on the spinal cord and cortical sensory systems. However, we also performed an exploratory, hypothesis-generating study to examine whether hyperalgesia was maintained after the reduction of inflammation in neonates receiving prolonged antibiotic treatment. In an independent sample of neonates who were treated with 5 days of antibiotics (referred to as the 'Neonatal Antibiotic Treatment Group'), the average CRP levels reduced from 26 mg/l (within 24 h from the presentation of risk factors) to 5 mg/l (within 96 h from the presentation of risk factors). However, when the magnitude of the noxious-evoked brain activity was compared with postnatal age-matched healthy neonates, with no evidence of infection, who were having a blood test for routine newborn screening ('Neonatal Antibiotic Control Group'), the magnitude of the brain activity was higher (magnitude of noxious-evoked brain activity: Neonatal Antibiotic Treatment Group: $n = 12$, mean = 1.21, SD = 1.02; Neonatal Antibiotic Control Group: $n = 8$, mean = 0.9, SD = 0.59; 34% increase, Fig. 3c).

As the neonates in each group were only matched in terms of age in this exploratory study, other factors such as prior pain experience or severity of illness were not necessarily comparable between groups, which limits the interpretation of this finding.

**Table 1 Participant demographics.**

|  | Neonatal Control Group | Neonatal Iinflammation Group | Neonatal Antibiotic Control Group | Neonatal Antibiotic Treatment Group |
|---|---|---|---|---|
| Number of neonates | 38 | 23 | 8 | 12 |
| Gestational age (GA) at birth (weeks) | 39.9 (37.8, 40.7) | 40 (39.2, 41) | 38.4 (36.7, 41) | 40.4 (39.9, 40.7) |
| Postmenstrual age (PMA) at the time of the study (weeks) | 40.1 (38, 40.9) | 40.1 (39.4, 41) | 39.1 (37.4, 41.8) | 41.1 (40.5, 41.4) |
| Postnatal age (PNA) at time of study (days) | 1 (1, 2) | 1 (1, 1) | 5 (5, 5) | 5 (4, 5) |
| C-reactive protein at time of study (mg/l) | 2.6 (1.6, 4.8) | 21.2 (11.1, 45.1) | - | 4 (3.1, 6.9) |
| Birthweight (g) | 3590 (3261, 4035) | 3818 (3405, 4158) | 3232.5 (2420, 4593) | 3803 (3704, 3950) |
| *Sex* |  |  |  |  |
| Male | 23 (61) | 13 (57) | 2 (25) | 8 (67) |
| Female | 15 (39) | 10 (43) | 6 (75) | 4 (33) |
| *Mode of delivery* |  |  |  |  |
| Normal vaginal delivery | 17 (45) | 6 (26.1) | 5 (62.5) | 3 (25) |
| Assisted breech vaginal | 0 (0) | 1 (4.3) | 0 (0) | 1 (8.3) |
| Assisted vaginal ventouse/forceps | 10 (26) | 10 (43.5) | 0 (0) | 2 (16.7) |
| Emergency C-section | 8 (21) | 4 (17.4) | 3 (37.5) | 6 (50) |
| Elective C-section | 3 (8) | 2 (8.7) | 0 (0) | 0 (0) |
| Apgar score at 1 min | 9 (7, 10) | 9 (7, 9) | 9 (8, 10) | 9 (9, 9) |
| Apgar score at 5 mins | 10 (9, 10) | 10 (10, 10) | 10 (10, 10) | 10 (10, 10) |
| Apgar score at 10 mins | 10 (10, 10)* | 10 (10, 10) | 10 (10, 10) | 10 (10, 10) |
| *Ventilation at time of study* |  |  |  |  |
| Self-ventilating in air (SVIA) | 35 (92) | 21 (91) | 8 (100) | 11 (91.7) |
| Low flow (LFT) | 1 (3) | 0 (0) | 0 (0) | 0 (0) |
| High flow (HFT) | 2 (5) | 2 (9) | 0 (0) | 1 (8.3) |
| Estimated cumulative prior pain exposure | 2 (1, 4) | 2 (1.5, 4) | 0 (0, 3) | 8 (4, 10) |

Values given are median (lower quartile, upper quartile) or number (%).
*Data not documented in clinical notes for one neonate.

Nevertheless, one interpretation of this exploratory observation is that hyperalgesia could outlive the acute inflammatory period that caused it, with neonatal inflammation potentially causing long-term changes in pain sensitivity. Our current experimental design does not allow for formal hypothesis-testing, but if this trend was to be confirmed in a future study, it would be consistent with observations made by other research groups in pre-clinical models[19,33,34]. It would also provide a potential underlying biological mechanism for what has been observed in epidemiological studies, which report that chronic pain is more likely in individuals who have experienced adverse events in childhood or adolescence[35]; this includes physical trauma and serious illness, all presumably associated with significant levels of inflammation. However, the effect observed in the exploratory study might not be larger than the differences observed due to random chance. Thus, an alternative interpretation of this observation is that early-life infection that has been treated with antibiotics results in no long-term consequences in noxious-evoked responses. An appropriately powered confirmatory study is warranted to test the hypothesis derived from this exploratory study. While CRP was used in this study as a marker of inflammation, there is value in complementing the current approach in future studies by exploring other factors such as elevated leucocyte count and other inflammatory markers commonly reported in the literature[10].

In conclusion, our results indicate that there is a connection between neonatal inflammation, spinal cord hypersensitivity, and cortical nociceptive processing in human infants. It represents a vital step towards translating what has become a sizable pre-clinical literature on neonatal inflammation and pain — which, until now, has been unmoored from clinical data. Future work,

involving carefully matched, longitudinal cohorts of neonates, will be required to further elucidate the potential long-term consequences of early-life neuro-immune interactions.

## Methods

**Participants.** A total of 65 term neonates were recruited for our hypothesis-testing study from the Newborn Care Unit and Maternity wards of the John Radcliffe Hospital (Oxford University Hospitals NHS Foundation Trust, Oxford, United Kingdom) between September 2014 and November 2019 (Supplementary Figure 2 shows the study profile). Data acquisition was completed in 61 studies. Participants in the Neonatal Control Group ($n = 38$) and Neonatal Inflammation Group ($n = 23$) were born between 36 and 42 weeks' gestation and were <72 h old at the time of the study (Table 1). All neonates in these groups presented with risk factors or clinical indicators of early-onset neonatal infection, required a blood test to quantify CRP levels and received the same regimen of intravenous antibiotics (commencing 18–24 h before the study, see[18] for guidelines that were in place during the study period). No positive blood cultures were recorded during the study period.

Researchers were blinded to the participants' clinical status at the time of the study and neonates were assigned to the Neonatal Control Group or Neonatal Inflammation Group after the study was complete based on clinical assessment, CRP levels within 24 h from the presentation of risk factors, other laboratory results and the clinical decision to continue with intravenous antibiotics to complete a minimum of a 5-day course (Fig. 1). Participants were grouped into the Neonatal Inflammation Group (CRP > 10 mg/l or other laboratory or clinical evidence of infection) or Neonatal Control Group (CRP < 10 mg/l and clinically asymptomatic). The threshold of 10 mg/l was selected because, in our institution, prophylactic antibiotic treatment was discontinued 36-hours after the first dose if the CRP was below 10 mg/l and there were no other clinical or laboratory signs of infection (Fig. 1).

The follow-up exploratory study included 20 participants (2 from the hypothesis-testing study cohort and 18 from an independent sample) recruited from the Newborn Care Unit and Maternity wards of the John Radcliffe Hospital, born between 36 and 42 weeks' gestation and between 4 and 6 days old at the time of the study. Neonates receiving a 5-day regimen of intravenous antibiotics due to suspected infection were studied during a blood test required to quantify CRP

**Table 2 Noxious-evoked spinal reflex withdrawal and brain activity in the Neonatal Inflammation Group and Neonatal Control Group.**

| | Neonatal Inflammation Group | | Neonatal Control Group | | Effect size estimation | | Unpaired two-sample t test | |
|---|---|---|---|---|---|---|---|---|
| | Mean | SD | Mean | SD | Difference in sample means (Δ) | 95%CI | t statistic | p value* |
| Reflex withdrawal (μV) | 42.1 | 30.9 | 30.2 | 20.6 | 11.9 | [1.32, ∞] | 1.74 | 0.048 |
| Brain activity (a.u.) | 1.04 | 0.68 | 0.64 | 0.52 | 0.41 | [0.13, ∞] | 2.4 | 0.022 |

*μV microvolts, a.u. arbitrary units, SD standard deviation. *one-sided p values corrected for multiple comparisons using Holm's method.*

levels 84–96 h after the first dose of antibiotics; participants with CRP < 10 mg/l at the time of the study were included in the Neonatal Antibiotic Treatment Group (n = 12) as they previously had raised CRP levels. Age-matched neonates with no previous evidence of infection or antibiotic treatment who required a blood test for routine newborn screening (newborn blood spot test and serum bilirubin) were included in the Neonatal Antibiotic Control Group (n = 8) (Table 1).

Neonates with HIE, congenital malformations or syndromes affecting the neurological system, receiving analgesics, or with a history of maternal substance abuse were not eligible to take part in the study. Participant demographics were recorded from the clinical notes and are presented in Table 1. Cumulative prior pain exposure was quantified as the total number of acute tissue-damaging procedures performed including heel lances, injections, and intravenous cannulation[36]. Ethical approval was obtained from the National Research Ethics Service, UK (reference: 12/SC/0447) and parental written informed consent was obtained before each participant was studied. The study was conducted in accordance with the standards set by the Declaration of Helsinki and Good Clinical Practice guidelines.

**Heel lance and tactile stimulus (non-noxious control)**. Neonates were only studied if a heel lance was necessary as part of their clinical care to assess their CRP levels or for routine screening. As standard practice, the comfort of the neonates was prioritised during the study with the use of swaddling and/or non-nutritive sucking. Lancing was performed on the heel with a mechanical BD Quikheel Premier Infant Lancet (Becton, Dickinson and Company, Franklin Lakes, NJ) with a penetration depth of 1.0 mm; following this, the foot was not squeezed for 30 s to exclude any effects of additional stimuli on the measured response. During each test occasion, the heel lace was preceded by a tactile stimulus (non-noxious control) where the mechanical lancet device was rotated 90° and activated so that the response to the tactile components of the stimulus was recorded without the lancet blade being in contact with the neonate's skin. Heel lances and non-noxious controls were automatically time-locked to the electrophysiological recordings using an event detection interface[37].

**Electrophysiological activity**. Participants' electrophysiological activity was acquired from DC to 400 Hz using a SynAmps RT 64-channel headbox and amplifiers (Compumedics Neuroscan). The activity was recorded using CURRY scan7 neuroimaging suite (Compumedics Neuroscan), with a sampling rate of 2000 Hz. Electroencephalography (EEG) was recorded from eight electrode sites (Cz, CPz, C3, C4, Oz, FCz, T3, T4), according to the modified international 10–20 system with reference at Fz and ground at Fpz. Disposable Ag/AgCl cup electrodes (Ambu Neuroline) were placed with conductive paste (Elefix EEG paste, Nihon Kohden) following a gentle clean of the scalp with preparation gel (Nuprep gel, D.O. Weaver and Co.) applied with a cotton bud. Surface electromyography (EMG) was recorded using bipolar EMG electrodes (Ambu Neuroline 700 solid gel surface electrodes) placed on the bicep femoris muscles from both legs.

EEG signals were filtered from 0.5–30 Hz with a notch filter at 50 Hz. Data were epoched from 500 ms before the stimulus to 1000 ms after and were baseline corrected to the pre-stimulus mean. Epochs were rejected if they contained gross movement artefact. Event-related potentials were analysed at the Cz electrode for all trials. The magnitude of noxious-evoked brain activity during the heel lance was obtained by projecting a previously validated template of noxious-evoked brain activity onto each individual trial 400–700 ms after stimulation[38]. Each individual trial was first Woody filtered to the template with a maximum jitter of ±100 ms in the time window of interest to account for individual differences in the latency to the response. Following rejections, a total of 51 participants were included in the noxious-evoked EEG data analysis (6/38 traces with artefact were rejected from the Neonatal Control Group and 4/23 traces—3 with artefact and 1 due to technical failure—were rejected from the Neonatal Inflammation Group, Supplementary Figure 2). The exploratory study EEG data were processed and analysed using the same methodology and 20 participants (Neonatal Antibiotic Treatment Group: n = 12, Neonatal Antibiotic Control Group: n = 8) were included in the final noxious-evoked EEG data analysis.

The origin and pain-related functions of the noxious-evoked potential fit by the template are difficult to assert given the lack of source localisation studies and comprehensive experimental designs in infant pain research that directly address

this question. However, we can tentatively draw some superficial similarities between the infant's late positive template potential, which is centrally distributed and maximal at Cz[38], and the adult noxious-evoked P2 potential, which is also a late positive potential centrally distributed and maximal at Cz[39–42]. This adult potential originates mainly from the mid-cingulate cortex[39,41,43], a brain region with pain-related nocifensive behavioural functions such as avoidance behaviour and body orientation to stimulus[43], which is also known to be active following noxious events in infants[44].

An early potential common to both noxious and tactile stimulation has been previously identified during a heel lance and a non-noxious control[29,38,45,46]. To characterise this early component in the EEG responses we used data from the Neonatal Control and Neonatal Inflammation Groups recorded during the background, heel lance, and non-noxious control stimuli. EEG signals were filtered from 0.5–30 Hz with a notch filter at 50 Hz. Data were epoched from 500 ms before the stimulus to 1000 ms after and were baseline corrected to the pre-stimulus mean. Epochs were rejected if they contained gross movement artefact (Supplementary Figure 2) and 2 further participants were rejected because background activity was not recorded. We included noxious-evoked EEG epochs from 49 neonates and tactile-evoked traces from 46 participants to characterise this early potential (Supplementary Figure 1). Event-related potentials were analysed at the Cz electrode for all trials. The morphology of the average raw EEG signal in the central electrode (Cz) was consistent with previous reports showing an early potential during both tactile and noxious-stimuli ~250 ms after the stimulus[29,38,45,46] (Supplementary Figure 1a). Heel lance and tactile epochs from all individual trials were combined and aligned with respect to the average of the data by Woody-filtering with a maximum shift of ±50 ms in the 0–700 ms interval after the stimulus onset. Clusters of timepoints where the combined noxious and tactile stimuli were significantly different from the background were identified using the nonparametric statistical analysis described by Maris and Oostenveld[47]. The cluster-based test statistic was calculated from 1000 random permutations of the data, and the threshold for cluster significance was set as the 97.5 percentile of the permuted data.

Heel lance and tactile stimulation evoked an activity cluster significantly different from background activity in the time window from 151–272 ms after the stimulus (p = 0.001, cluster-corrected nonparametric test, Supplementary Figure 1b). Principal component analysis (PCA) was applied to the EEG epochs in the time window 100–350 ms to capture the whole waveform and 94% of the variance was captured by the first four principal components (PCs). The first PC morphology was comparable to that identified in previous studies[29,38,45,46] (Supplementary Figure 1c) and its weights were significantly higher following the heel lance (repeated measures ANOVA with multiple comparisons p = 8.2e-06) and the tactile stimulus (p = 0.0008, p values corrected using Bonferroni's method) compared to background brain activity (Supplementary Figure 1d). The weights of this PC for the tactile-evoked EEG (Neonatal Control Group n = 29, Neonatal Inflammation Group n = 19, Supplementary Figure 2) represent the magnitude of the early potential in response to a tactile stimulus.

EMG signals were filtered 10–500 Hz, with a notch filter at 50 Hz and harmonics, and rectified. Epochs were extracted from 2 s before to 4 s after the stimulus. Individual epochs were rejected due to movement artefact in the baseline period. The data was split into 250 ms windows and the RMS of the reflex signal was calculated in each window[48]. The average RMS across the first four windows after the stimulus (first second after stimulation) was calculated as the magnitude of the reflex withdrawal. Following artefact removal, a total of 57 participants were included in the final noxious and tactile-evoked EMG data analysis (2/38 traces with movement artefact were rejected from the Neonatal Control Group and 2/23 traces –1 with artifact and 1 due to technical failure—were rejected from the Neonatal Inflammation Group, Supplementary Figure 2).

**Premature Infant Pain Profile-Revised**. A PIPP-R score, which combines behavioural and physiological measures and contextual factors[49], was calculated in response to the stimuli for all the participants in the hypothesis-testing study. The neonates' behavioural responses to the stimuli were evaluated by recording their facial expressions with a handheld camera 15 s before and 30 s after the heel lance and tactile procedures. A LED flash was activated by the researcher simultaneously with each stimulation and used as a marker for the time of stimulation. The recording of each procedure was retrospectively analysed by researchers blinded to

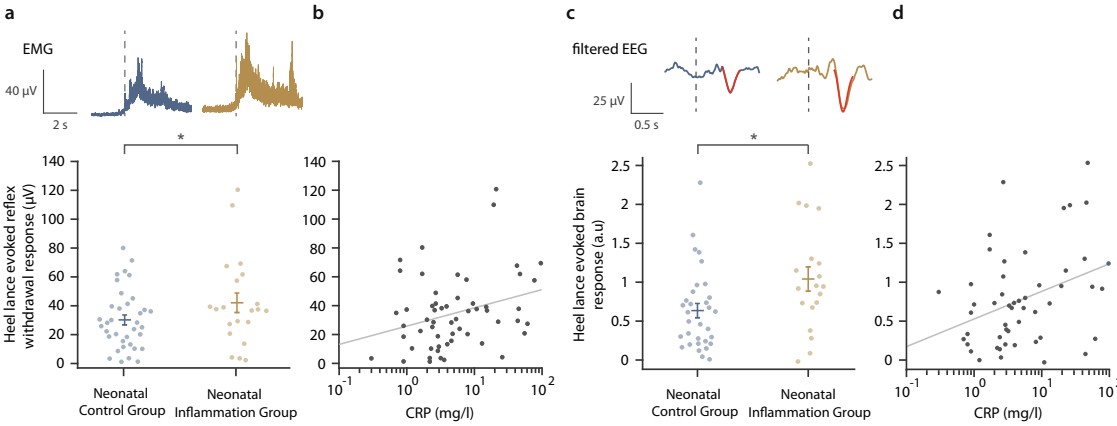

**Fig. 2 Inflammation is associated with increased spinal cord excitability and hyperalgesia during a noxious procedure in neonates. a** (Top) Average electromyography (EMG) traces during a clinically required heel lance for participants in the Neonatal Control Group (blue, $n = 36$) and Neonatal Inflammation Group (gold, $n = 21$, CRP > 10 mg/l or presented evidence of infection). (Bottom) Root mean square (RMS) of the reflex withdrawal in the limb ipsilateral to the stimulus site in the two groups. The difference between groups was assessed using a two-sample, one-sided $t$ test, $t = 1.74$, *$p = 0.048$ (corrected using Holm's method). **b** Relationship between the CRP level at the time of the study and the magnitude of the reflex withdrawal following a clinically required heel lance ($n = 57$, Pearson correlation $r = 0.28$, one-sided $p = 0.028$). **c** (Top) Group average Woody filtered electroencephalography (EEG) traces in response to the clinically required heel lance; Neonatal Control Group ($n = 32$) and Neonatal Inflammation Group ($n = 19$). The template of noxious-evoked brain activity[38] is shown overlaid in red. (Bottom) Magnitude of the noxious-evoked brain activity following heel lancing in the two groups. The difference between groups was assessed using a two-sample, one-sided $t$ test, $t = 2.4$, *$p = 0.022$ (corrected using Holm's method). **d** Relationship between the CRP level at the time of the study and the magnitude of the noxious-evoked brain activity following a clinically required heel lance ($n = 51$, Pearson correlation $r = 0.31$, one-sided $p = 0.023$). **a, c** dashed lines indicate the point of stimulation, and error bars indicate mean ± standard error, in **b** and **d**, solid line indicates the line of best fit. a.u: arbitrary units. Source data are provided as a Source Data file.

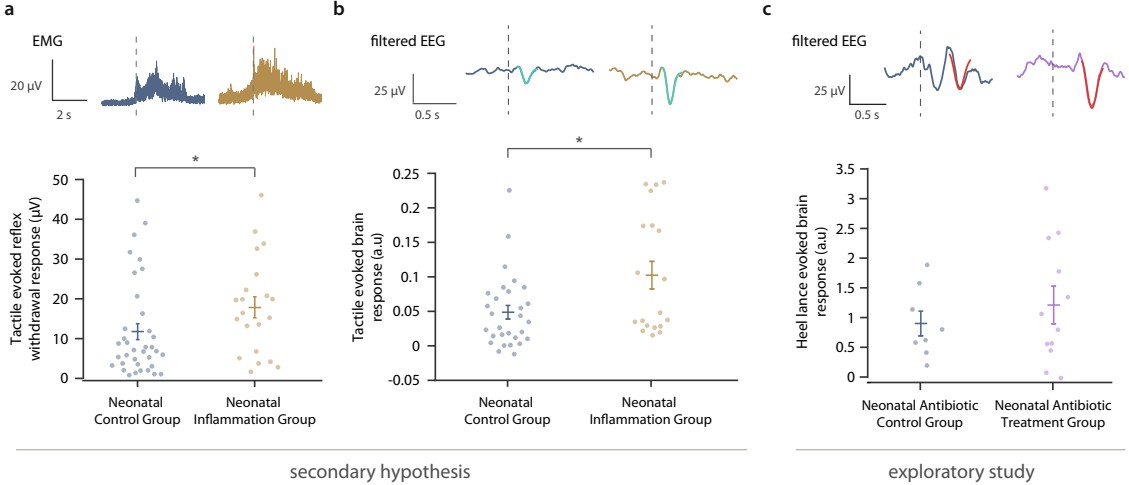

**Fig. 3 Neonatal inflammation is associated with increased spinal cord excitability and arousal-related brain activity during a tactile stimulus, and noxious-evoked brain activity results from an exploratory study. a** (Top) Average electromyography (EMG) traces during a tactile stimulus for participants in the Neonatal Control Group (blue, $n = 36$) and Neonatal Inflammation Group (gold, $n = 21$, CRP > 10 mg/l or presented evidence of infection). (Bottom) Root mean square (RMS) of the reflex withdrawal in the limb ipsilateral to the tactile stimulus site in the two groups. The difference between groups was assessed using a two-sample, one-sided $t$ test; $t = 1.85$, *$p = 0.037$. **b** (Top) Group average (Woody) filtered electroencephalography (EEG) traces in response to a tactile stimulus; Neonatal Control Group ($n = 29$) and Neonatal Inflammation Group ($n = 19$). The evoked potential is shown overlaid in teal. (Bottom) Magnitude of the brain activity following a tactile stimulus in the two groups. The difference between groups was assessed using a two-sample, one-sided $t$ test; $t = 2.66$, *$p = 0.007$. **c** Exploratory study: (Top) Group average (Woody) filtered EEG traces in response to the clinically required heel lance; Neonatal Antibiotic Control Group (blue, $n = 8$) and Neonatal Antibiotic Treatment Group (purple, $n = 12$). The template of noxious-evoked brain activity[38] is shown overlaid in red. (Bottom) Magnitude of the noxious-evoked brain activity following heel lance in the two groups. In **a–c** dashed lines indicate the point of stimulation and error bars indicate mean ± standard error. a.u: arbitrary units. Source data are provided as a Source Data file.

the stimulus type to estimate the facial expressions' scores of the PIPP-R by individually assessing the duration of brow bulge, eye squeeze, and nasolabial furrow expressions over the 30 s after the stimulus. Oxygen saturation and heart rate were acquired with a pulse oximeter and ECG using an IntelliVue MX800 Philips patient monitor and vital signs were continuously downloaded from the monitor using ixTrend software (ixitos GmbH, Germany). The time of stimulation

was manually annotated on the computer recording during the study by a researcher. The heart rate and oxygen saturation scores were determined as the difference between the baseline averages in the 15 s pre-stimulus and the maximum and minimum levels respectively in the 30 s post-stimulation. The maximum PIPP-R score is 21 with 0 indicating no pain, 1–6 mild pain, 7–12 moderate pain, and >13 severe pain[50]. Owing to technical issues (1/38: Neonatal Control Group and 1/

23: Neonatal Inflammation Group) and missing physiological data (3/38: Neonatal Control Group and 1/23: Neonatal Inflammation Group), a total of 55 neonates (Neonatal Control Group: $n = 34$, Neonatal Inflammation Group: $n = 21$) were included in the clinical pain scores (PIPP-R) analysis (Supplementary Figure 2).

**Sample size calculation and statistical analysis**. Here we present two studies: the first study is a hypothesis-testing study that tests two hypotheses and the second is an exploratory study. The principal hypothesis was that neonatal inflammation causes increased spinal cord excitability and hyperalgesia in response to noxious stimulation. The second hypothesis was that neonatal inflammation causes increased spinal cord excitability and arousal-related brain activity in response to tactile stimulation. A power calculation was performed to determine the sample size required to test the principal hypothesis.

The outcome measures of this study were the noxious-evoked reflex withdrawal and brain activity in response to a heel lance in the Neonatal Inflammation Group and Neonatal Control Group. A 70% increase in the Neonatal Inflammation Group was considered to be clinically significant as this effect has been reported for inflammation-induced hyperalgesia in adults[51]. The mean (SD) brain activity evoked by a heel lance in a cohort of healthy term infants is 0.72 (0.69) (data published in refs. [52] and [53]). A sample size of 56 neonates (2:1 Neonatal Control Group to Neonatal Inflammation Group allocation ratio) would be required to observe a 70% increase in noxious-evoked brain activity with a two-sample t-test (80% power and a one-sided 5% significance level). We allowed for a total sample size of 65 neonates to account for 15% of losses due to technical failures or clinical ineligibility (it was expected that after enrolment some infants would require a recannulation for antibiotic administration therefore precluding the need to perform a heel lance). We also calculated the sample size that would be required to achieve adequate power to observe a significant difference in reflex withdrawal activity between the two groups. The mean (SD) RMS of the reflex withdrawal in a cohort of healthy term infants is 23.3 (17.7) (data published in refs. [52] and [53]). Assigning the same assumptions as for the noxious-evoked brain activity a lower sample size of 36 neonates would be required for this measure. Therefore, a sample size of 65 ensured that adequate power could be achieved for both outcome measures.

CRP is routinely used during Early-Onset Neonatal Infection screening in the clinic and was used to split our participants between the Neonatal Control Group and Neonatal Inflammation Group. Due to the limited sample sizes available to us, we did not consider the relationship between EEG or EMG activity and the impact of other markers of inflammation. Power calculations were performed in G*Power v3.1[54]. Sample size calculations were not performed for the exploratory study as no hypotheses were being tested.

Statistical analysis was performed in MATLAB_R2020 (MathWorks) and R version 3.6.3 (The R Project for Statistical Computing). Group differences in noxious-evoked brain activity, tactile-evoked brain activity, reflex withdrawal, and clinical pain scores were assessed using unpaired two-sample t-tests. Statistical significance (one-sided alpha <0.05) was assessed non-parametrically via permutation testing with 10,000 permutations using the PALM (permutation analysis of linear models) toolbox[55]. One-sided tests were used to reflect the directional hypotheses presented in this study. The effect size estimations were calculated using the DABEST web application (https://www.estimationstats.com)[56], the 95% confidence interval of the mean difference was calculated with bootstrapping, by taking 5000 samples with replacement. Linear associations between noxious-evoked responses and CRP levels were assessed using Pearson correlation tests and statistical significance (one-sided alpha <0.05) was assessed non-parametrically via permutation testing with 10,000 permutations using PALM. Holm's method[57] was used to correct p values for multiple comparisons across the principal hypothesis analysis (two comparisons: noxious-evoked brain activity between groups and noxious-evoked reflex withdrawal between groups). P values for secondary hypothesis tests are presented without correction for multiple comparisons as the purpose of these secondary analyses was to support and provide greater insight into the primary results. Statistical details for each analysis (number of participants, $t$, $r$, $\Delta$, CI, and $p$ values) are presented in the text.

**Reporting summary**. Further information on research design is available in the Nature Research Reporting Summary linked to this article.

## Data availability

The data that support the findings of this study are available from the corresponding author (rebeccah.slater@paediatrics.ox.ac.uk) within three months upon request. Due to ethical restrictions, we consider it appropriate to monitor the access and usage of the data as it includes highly sensitive information. Source data to produce Figs. 2, 3, and Supplementary Figure 1 are provided in this paper.

## Code availability

The code used to quantify the magnitude of noxious- and tactile-evoked brain activity is available from[38] (Supplementary Material – Data file S2) https://www.science.org/doi/10.

1126/scitranslmed.aah6122#supplementary-materials. The template of noxious-evoked brain activity is available from[38] (Supplementary Material – Data file S1) https://www.science.org/doi/10.1126/scitranslmed.aah6122#supplementary-materials, and the early-event potential is provided with this publication as a source data file.

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

## Acknowledgements

The study was funded by the Wellcome Trust via a Senior Wellcome Fellowship awarded to Rebeccah Slater (Grant number 207457/Z/17/Z) and a Bliss research grant. Caroline Hartley is a Wellcome Trust/Royal Society Sir Henry Dale Fellow (213486/Z/18/Z). We would like to thank the newborn participants and their parents for taking part in this study.

## Author contributions

M.M.C. data curation, formal analysis, validation, investigation, visualisation, methodology, project administration, writing—original draft, writing—review, and editing. G.G.: conceptualisation, data curation, formal analysis, investigation, writing—review, and editing. F.A., R.E.F., A.H., G.S.M.: investigation, writing—review and editing, clinical oversight of the studies. A.G., D.G., M.vd.V.: data curation, investigation, writing—review and editing. L.B.: formal analysis, validation, writing—original draft, writing—review, and editing. E.A.: investigation, resources, writing—review and editing, clinical oversight of the studies, supervision. A.B.: investigation, visualisation, project administration, writing—review and editing, supervision. F.D.: validation, visualisation, writing—original draft, writing—review, and editing. C.H.: data curation, formal analysis, validation, investigation, visualisation, methodology, writing—original draft, writing—review, and editing, supervision. R.S.: conceptualisation, formal analysis, investigation, resources, validation, visualisation, methodology, project administration, writing—original draft, writing—review and editing, supervision, funding acquisition.

## Competing interests

The authors declare no competing interests.
