## [Peer Review File · Nature Communications]

Reviewers' Comments:

Reviewer #1:

Remarks to the Author:

The question of a possible link between inflammation and nociceptive sensitivity is an interesting one, and certainly worth exploring. The authors have admirably performed a study on live human subjects at an early age, which is not easy to do. However, I have some significant concerns about this manuscript. Specific comments and recommendations for improvement are as follows.

1. The p values presented on page 3 of the manuscript ($p = 0.048, 0.036, 0.035$) are not very convincing for statistically significant changes, as they are barely below the traditional 0.05 threshold. In Figure 2 and Figure 3, the scatterplots show a great deal of overlap in the distribution of the data points, also raising concerns about how meaningful these differences are
2. The quantification of the EMG and EEG signals is described in sufficient detail in the methods but it is not clear why they are being interpreted as they are. These signals provide at best a limited insight into what is happening in the nervous system during the timepoints being measured. To document spinal cord excitability, it would have been much more meaningful to have recorded more clearly interpretable neurophysiologic data such as somatosensory evoked potentials. It would have been immensely helpful to have enlisted the aid of a pediatric neurologist or neurophysiologist to help design and interpret the EMG and EEG data.
3. It is problematic to use CRP levels and empiric use of antibiotics as proxies for inflammation. I understand that confirmed bacterial infections and other documented inflammatory states are less common than before, which is a good development, but it would have been better to define inflammation more clearly, such as a positive blood culture or supplementing the CRP criterion with an elevated leukocyte count.
4. In the methods, it states that "Neonates with neurological conditions" were excluded. Does this mean that neonates with meningoencephalitis were excluded?
5. Though the EEG methods are presented in detail, it is not clear which of the 8 electrodes were used to calculate the data. It states that "Event related potentials were analyzed at the Cz electrode for all trials." Each electrode must be compared with another to derive a meaningful signal. What was the Cz electrode being compared to?
6. It is not clear where the electrodes were placed for the EMG. Were these surface electrodes or needle electrodes? Surface electrodes are subject to significant artifact and it is difficult to know what differences in surface EMG activity mean.

Reviewer #2:

Remarks to the Author:

Aim: This study evaluates the relationship between neonatal inflammation and pain in newborn human infants. The authors anticipate that there will be a relationship between the immune and nociceptive systems, similar to animal studies, although to date the relationship has been unclear and not explored in humans. The research question is highly relevant and important given that approximately 13% of live births have suspected neonatal infection and the long-term effects of pain during this critical stage of development have the potential to be deleterious long-term.

Methods: The investigators studied matched cohorts of full-term neonates with suspected infection (based on C-reactive protein (CRP) levels) grouped as those with $CRP > 10$ mg/l being considered as those with infection (Neonatal Inflammation Group) and those with $CRP < 10$ mg/l as those without infection (Neonatal Control Group).

They hypothesized that neonatal inflammation caused increased spinal cord excitability and increased cortical brain activity in response to (i) a noxious heel lance and (ii) a non-noxious tactile stimulation to the heel. These hypotheses were tested measuring EMG and EEG outputs as

well as a clinical validated infant pain scale (PIPP-R). The authors also conducted an exploratory hypotheses-generating study with an additional 20 infants who were treated with 5 days of antibiotics to determine if the magnitude of brain activity was higher than those who were healthy postnatal matched neonates with no evidence of infection.

Results: In response to heel lance, infants in the Neonatal Inflammation Group had greater ECG and EEG responses than those in the Neonatal Control Group. These results are somewhat difficult to follow – based on how they are stated in the text – perhaps a table would clarify. The results of the non-noxious tactile stimulation are more clearly stated. Although there was statistical significance between the groups, given that the SD were large, some explanation of the meaning of these scores would be useful. There was no difference between the groups using the clinical infant pain scale. This lack of difference was explained by the fact that inflammation suppresses behavioural activity and induces lethargy, thus suggesting that clinical measurement of pain may not be appropriate for infants with pathological conditions. This conclusion seems premature given the small sample size in this single study. In the exploratory study, although there were no statistically significant difference in outcomes, the authors reported that the magnitude of brain activity was higher in the infection group, and this could possibly be interpreted as the inflammation having the potential to cause long-term changes in pain sensitivity. Again, given the small sample size and the exploratory nature of this study, this conclusion seems premature. There were no data presented on clinical scores of pain for this exploratory part of the study and it was not clear if this was done or not. Information on how the antibiotics were delivered (e.g. intravenously or orally) was also not specified.

The methods were well described and included clear inclusion/ exclusion criteria. A sample size calculation was included to ensure an adequate sample size for the main hypothesis testing – none was provided for the exploratory study thus we do not know if the sample size was adequate for this exploratory work. A few infants were lost, due to technical issues; leaving 55 neonates for the analyses – this needs to be further clarified in Table 1. In terms of description of the PIPP-R scores, it is important to state how the 2 contextual variables were calculated and whether these were included in the total PIPP-R scores given the revised scoring instructions. The PIPP-R scores for full term infants are also calculated out of a total of 18 (not 21). Statistical analyses appears to be appropriate.

Overall: This is an important first study, in looking at the relationship between inflammation and pain in human neonates thus extending our knowledge from animal studies. The methods are well conceived, although there may not be an adequate sample size for the exploratory study. Some details require further clarification as outlined above. The results should be reported without overgeneralizing. The conclusion that an adequately powered prospective study is the next logical step in accurate.

Reviewer #3:

Remarks to the Author:

The Slater group has for a decade or more made substantial contributions to the infant pain literature. They have pioneered EEG, EMG, behavior and brain imaging work in neonates, linking those data closely in some cases to the basic science literature. This is an additional study showing that ongoing infection with antibiotics increases spinal and brain reactivity to noxious and tactile stimuli. The data are solid as are the methods. However, the interpretation of the data is problematic.

First, because most the the preclinical and some of the clinical data on infant infection looks at longer term outcomes that outlast the infection it should be clearly emphasized that the infection is concurrent with the testing. The abstract and intro are accurate but the introduction emphasizes long-term outcomes (which indeed is the bulk of the literature). For example, the first paragraph states outcomes from early infection that occur later in life - asthma, type-1 diabetes, later behavioral, motor and cognitive behavior. The basic science papers cited, especially those of Zouikr, are early injury and later outcomes. Indeed ref. 15, showed that 2 days after LPS there were no changes in a formalin challenge and that it took >2 weeks after LPS challenge to show

changes in SDH.

Also, the fact that the two independent variables are infection and concurrent antibiotic treatment, the title and abstract need to include that. Are there prior data on the effects of antibiotics on pain independent of their anti-bacterial effects. That needs some discussion and is a clinically mandated confound.

Although it is clear that tactile and noxious stimuli produce an exaggerated spinal and brain response, the PIPPS-R data are not different between groups. The argument that "sickness syndrome" had an effect that reduced the validity of the PIPPS-R data certainly may be true. Would that also be expected to alter the EMG? Are there other data showing a discrepancy between PIPPS and biological measurements of pain. Are there data on the PIPPS in other infection studies? To this reviewer, the authors have balanced the biological and behavioral outcomes well but more discussion of other PIPPS data might be clarifying.

The hypothesis explorative study, 5 days after treatment or not with antibiotics, shows NO difference despite the statement that evoked brain activity was higher, albeit not significantly. The authors conclude "the hyperalgesia could outlive the acute inflammatory period". This statement is problematic and implies to this reader, a trend towards significance. But statistical analysis given the mean, SD and N of each group showed a p-value of ~ 0.45 , so clearly the two groups did NOT differ. Indeed the proper argument given these preliminary data is more reassuring - early life treated infection (or antibiotic treatment) has no long term consequence for noxious evoked brain activity.

Minor point - a bit of discussion on the possible effects of comforting the infant with swaddling and/or non-nutritive sucking. Perhaps the comforting affects the PIPP-R data?

Thus, my major critique is not of the data or methods but the slant put on those data in their interpretation. One could easily argue that infection+antibiotics, although concurrently altering EMG and EEG, has no behavioral consequence and that the altered EEG is transient at best. This is a different take on the data than that of the authors, but one that I think is equally valid.

Reviewer #1:

The question of a possible link between inflammation and nociceptive sensitivity is an interesting one, and certainly worth exploring. The authors have admirably performed a study on live human subjects at an early age, which is not easy to do. However, I have some significant concerns about this manuscript. Specific comments and recommendations for improvement are as follows.

We thank the reviewer for their detailed review and comments.

1. The p values presented on page 3 of the manuscript ($p = 0.048, 0.036, 0.035$) are not very convincing for statistically significant changes, as they are barely below the traditional 0.05 threshold. In Figure 2 and Figure 3, the scatterplots show a great deal of overlap in the distribution of the data points, also raising concerns about how meaningful these differences are

We acknowledge that the group comparisons (Neonatal Control group vs Neonatal Inflammation Group) p values presented alone would represent a modest effect of the condition of interest on the noxious-evoked activity. However, it is worth noting that CRP level is a continuous variable, which is dichotomised for purposes of clinical decision making. In our study, we used the cut off of $CRP > 10 \text{ mg/l}$ to define the two groups because this threshold drives the clinical diagnosis of suspected Early Onset Neonatal Infection (EONI) in the clinic, and it was worth directly assessing this group comparison. However, this pragmatic clinically-motivated dichotomisation of a continuous variable leads to loss of power due to close overlap in distributions of data points (Altman & Royston, 2006). Thus, the overlap in distributions is primarily due to the continuous nature of CRP level. We also directly assessed CRP level correlations, as presented in Figure 2b and 2d. These correlations between CRP level and reflex withdrawal responses (Figure 2b) and CRP level and noxious-evoked brain activity (Figure 2d) were statistically significant. Taken together, the combined evidence of significant correlations with CRP level and significant differences between groups (when CRP level is dichotomised at the clinical decision-making threshold of 10 mg/l), suggest these effects are potentially of clinical value, despite their modest p-values.

We have edited the results to highlight this point and now provide a clearer link between the group comparisons and the correlations. The following text has been added.

“Due to the continuous nature of the CRP level variable, we assessed correlations between CRP level and EMG and EEG responses, and both the spinal cord mediated reflex withdrawal activity and noxious-evoked brain activity significantly correlated with the level of CRP in the blood sample (reflex withdrawal activity: $n=57$, Pearson correlation $r=0.28$, $p=0.035$, noxious-evoked brain activity: $n=51$, $r=0.31$, $p=0.028$, Figure 2). Taken together, the combined evidence of significant correlations with CRP level and significant differences between groups (when CRP level is dichotomised at the clinical decision-making threshold of 10 mg/l), suggest these effects are potentially of clinical value, despite their modest p-values” – page 4, lines 96-104.

2. The quantification of the EMG and EEG signals is described in sufficient detail in the methods but it is not clear why they are being interpreted as they are. These signals provide at best a limited insight into what is happening in the nervous system during the timepoints being measured. To document spinal cord excitability, it would have been much more meaningful to have recorded more clearly interpretable neurophysiologic data such as somatosensory evoked potentials. It would have been immensely helpful to have enlisted the aid of a pediatric neurologist or neurophysiologist to help design and interpret the EMG and EEG data.

While we fully agree that both EMG and EEG signals provide “at best a limited insight into what is happening in the nervous system during the timepoints being measured”, we do not consider that the use of somatosensory evoked potentials (SEPs) would overcome these methodological limitations or would be an appropriate alternative in a neonatal study primarily focussed on the relationship between inflammation and pain. The noxious stimulus used in these studies was a clinically-required heel lance, as opposed to

experimentally induced SEPs (which cannot be applied to neonates at an intensity that would be considered noxious).

Due to the immense challenges, pragmatic and ethical, in assessing pain in newborns, we employed bespoke methods that our group and others have been establishing and validating for over a decade (Goksan et al., 2015; Hartley et al., 2017; Slater et al., 2006, 2010). The measurement of noxious-evoked responses such as limb withdrawal (assessed with EMG) and noxious-evoked potentials (assessed with EEG) is appropriate to test our hypothesis “We hypothesised that neonatal inflammation causes increased spinal cord excitability and increased cortical brain activity” (page 4, lines 78-79) and allowed us to directly establish how inflammation impacts neonatal responses to painful events. Our interpretation of our EEG and EMG signals in this manuscript are fully consistent with interpretations in previous clinical studies (Cobo et al., 2021; Gursul et al., 2018; Hartley et al., 2015; Kasser et al., 2019) and in a clinical trial (Hartley et al., 2018).

Finally, we fully agree that there is great value in enlisting the aid of paediatric neurologists or neurophysiologists, which is why researchers with these professions feature in our author list.

3. It is problematic to use CRP levels and empiric use of antibiotics as proxies for inflammation. I understand that confirmed bacterial infections and other documented inflammatory states are less common than before, which is a good development, but it would have been better to define inflammation more clearly, such as a positive blood culture or supplementing the CRP criterion with an elevated leukocyte count.

We agree that the use of CRP levels as the sole indicator of inflammation is a potential limitation of our study and that the measurement of other inflammatory markers would have allowed us to better define cases of infection. A range of inflammation markers have been used in the pre-clinical and adult literature, mostly quantifying pro-inflammatory cytokines closely associated with hyperalgesia (Barr & Hunter, 2014). CRP is an acute-phase inflammatory protein which exhibits elevated expression during inflammatory conditions. It has been traditionally used as a marker of infection and more recent evidence links CRP with specific inflammatory processes dependent on the complement pathway, nitric oxide release and the production of cytokines particularly IL-6 and TNF (Sproston & Ashworth, 2018).

Here, we used CRP as an inflammation marker because it is routinely used in the hospital where the study was conducted to drive the clinical diagnosis of EONI (National Institute for Health and Care Excellence (NICE), 2012). Given the relatively small sample sizes, we did not consider the impact of other markers of inflammation nor the relationship between EEG or EMG activity, which would have considerably increased the required sample size. Moreover, the measurement of additional markers of inflammation could have potentially required larger volumes of blood than those taken for clinical purposes, which may not have been appropriate in this first study.

We appreciate that it is important to clarify that in the cohort of neonates included in our study there were no positive blood cultures. As the reviewer mentioned it is very rare to have cases of confirmed infection by positive blood cultures in our geographical region. We have also clarified the importance of considering additional markers of inflammation in the manuscript:

“No positive blood cultures were recorded during the study period.” – page 7, lines 224-225.

“While CRP was used in this study as a marker of inflammation, there is value in complementing the current approach in future studies by exploring other factors such as elevated leukocyte count and other inflammatory markers commonly reported in the literature [10].” – page 7, lines 198-200.

4. In the methods, it states that "Neonates with neurological conditions" were excluded. Does this mean that neonates with meningoencephalitis were excluded?

Neonates with meningoencephalitis were excluded from our study. We have added the specific exclusion criteria to the Methods:

"Neonates with HIE, congenital malformations or syndromes affecting the neurological system, receiving analgesics or with a history of maternal substance abuse were not eligible to take part in the study." – page 8, lines 256-258.

5. Though the EEG methods are presented in detail, it is not clear which of the 8 electrodes were used to calculate the data. It states that "Event related potentials were analyzed at the Cz electrode for all trials." Each electrode must be compared with another to derive a meaningful signal. What was the Cz electrode being compared to?

We referenced all the electrodes to Fz and we used activity from Cz to perform the analysis. As mentioned in the 'Electrophysiological activity' section of the Methods, "Electroencephalography (EEG) was recorded from eight electrode sites (Cz, CPz, C3, C4, Oz, FCz, T3, T4), according to the modified international 10-20 system **with reference at Fz** and ground at Fpz." This is thus a referential montage with Fz as the reference, so all eight recording electrodes (including Cz) are compared to the Fz reference electrode.

6. It is not clear where the electrodes were placed for the EMG. Were these surface electrodes or needle electrodes? Surface electrodes are subject to significant artifact and it is difficult to know what differences in surface EMG activity mean.

As stated in the 'Electrophysiological activity' section of the Methods, "**Surface electromyography (EMG)** was recorded using bipolar EMG electrodes (Ambu Neuroline 700 solid gel surface electrodes) **placed on the bicep femoris muscles from both legs.**" While the use of needle electrodes would result in reduced artifact, it would not have been possible to use needle electrodes due to practical and ethical limitations.

Changes in surface EMG activity have been used by different research groups to describe the postnatal development of cutaneous flexion reflexes in and human infants (Andrews & Fitzgerald, 1994; Cornelissen et al., 2013; Hartley et al., 2016; Kühne, Stützer, Roth, & Welzing, 2012). In the context of pain assessment, reflex withdrawal recorded with surface EMG electrodes has been used to investigate sensory processing and pain thresholds in infants (Andrews & Fitzgerald, 1999). Its amplitude correlates with stimulus intensity, it is associated with the magnitude of noxious evoked brain activity (Hartley et al., 2015), and it has been demonstrated that the RMS measure used in this manuscript allows discrimination between noxious and non-noxious stimuli (van der Vaart et al., 2019).

Reviewer #2:

Aim: This study evaluates the relationship between neonatal inflammation and pain in newborn human infants. The authors anticipate that there will be a relationship between the immune and nociceptive systems, similar to animal studies, although to date the relationship has been unclear and not explored in humans. The research question is highly relevant and important given that approximately 13% of live births have suspected neonatal infection and the long-term effects of pain during this critical stage of development have the potential to be deleterious long-term.

Methods: The investigators studied matched cohorts of full-term neonates with suspected infection (based on C-reactive protein (CRP) levels) grouped as those with CRP >10 mg/l being considered as those with infection (Neonatal Inflammation Group) and those with CRP <10mg/l as those without infection (Neonatal Control Group).

They hypothesized that neonatal inflammation caused increased spinal cord excitability and increased cortical brain activity in response to (i) a noxious heel lance and (ii) a non-noxious tactile stimulation to the heel. These hypotheses were tested measuring EMG and EEG outputs as well as a clinical validated infant pain scale (PIPP-R). The authors also conducted an exploratory hypotheses-generating study with an additional 20 infants who were treated with 5 days of antibiotics to determine if the magnitude of brain activity was higher than those who were healthy postnatal matched neonates with no evidence of infection.

We thank the reviewer for their valuable comments.

Results: In response to heel lance, infants in the Neonatal Inflammation Group had greater ECG and EEG responses than those in the Neonatal Control Group. These results are somewhat difficult to follow – based on how they are stated in the text – perhaps a table would clarify. The results of the non-noxious tactile stimulation are more clearly stated.

As suggested by the reviewer, we have added the following table to summarise the key noxious-evoked spinal reflex withdrawal and brain activity results:

Table 1. Noxious-evoked spinal reflex withdrawal and brain activity in the Neonatal Inflammation Group and Neonatal Control Group.

	Neonatal Inflammation Group		Neonatal Control Group		Unpaired two sample t-test	
	Mean	SD	Mean	SD	t-statistic	p-value
Reflex withdrawal (μV)	42.1	30.9	30.2	20.6	1.74	0.048
Brain activity (a.u.)	1.04	0.68	0.64	0.52	2.4	0.036

μV = microvolts; a.u. = arbitrary units; SD = standard deviation.

Page 22, lines 657-659

Although there was statistical significance between the groups, given that the SD were large, some explanation of the meaning of these scores would be useful.

CRP level is a continuous variable, which is dichotomised for purposes of clinical decision making. This pragmatic clinically-motivated dichotomisation of a continuous variable leads to loss of power and overlapping distributions of data points (Altman & Royston, 2006). The relatively large standard deviations, that noticeably overlap between the two groups, are primarily due to the continuous nature of CRP level. However, due to the clinical utility of the dichotomisation of CRP level, it was worth directly assessing this group comparison. In our study, we used the cut off of CRP > 10 mg/l to define the two groups because this threshold drives the clinical diagnosis of suspected Early Onset Neonatal Infection (EONI) in the clinic. Due to the continuous nature of CRP level, we also directly assessed CRP level correlations. Taken together, the combined evidence of significant correlations with CRP level and significant difference between groups (when CRP level is dichotomised at the clinical decision-making threshold of 10 mg/l), suggest these effects are potentially of clinical value, despite large standard deviations. We have edited the results to highlight this point.

“Due to the continuous nature of the CRP level variable, we assessed correlations between CRP level and EMG and EEG responses, and both the spinal cord mediated reflex withdrawal activity and noxious-evoked brain activity significantly correlated with the level of CRP in the blood sample (reflex withdrawal activity: n=57, Pearson correlation r=0.28, p=0.035, noxious-evoked brain activity: n=51, r=0.31, p=0.028, Figure 2). Taken together, the combined evidence of significant correlations with CRP level and significant differences

between groups (when CRP level is dichotomised at the clinical decision-making threshold of 10 mg/l), suggest these effects are potentially of clinical value, despite their modest p-values” – page 4, lines 96-104.

There was no difference between the groups using the clinical infant pain scale. This lack of difference was explained by the fact that inflammation suppresses behavioural activity and induces lethargy, thus suggesting that clinical measurement of pain may not be appropriate for infants with pathological conditions. This conclusion seems premature given the small sample size in this single study.

We agree that our proposed interpretation of the PIPP-R score results may be premature and that we could be underpowered to observe an effect of inflammation on the clinical pain scores. We have edited page 5 to provide a more nuanced discussion of this point.

“While it is possible that we are underpowered to observe an effect on clinical pain scores (given the subjective nature of PIPP-R assessments in contrast to objective EMG/EEG assessments), our observation is consistent with clinical observations in adults and neonates, as well as data from animal studies, which demonstrate that inflammation suppresses behavioural activity, causing listlessness and lethargy [24,25]” - page 5, lines 122-126.

“Further research is warranted to establish how the presence of neonatal inflammation impacts clinical pain scores and the degree to which subjective observational assessment of infant pain-related behaviours might influence pain scoring.” - page 5, lines 134-137.

In the exploratory study, although there were no statistically significant difference in outcomes, the authors reported that the magnitude of brain activity was higher in the infection group, and this could possibly be interpreted as the inflammation having the potential to cause long-term changes in pain sensitivity. Again, given the small sample size and the exploratory nature of this study, this conclusion seems premature.

We thank the reviewer for this comment, and we acknowledge that the phrasing was unclear when we reported the exploratory study results. Our paper presents two studies: the first is a confirmatory study (hypothesis-testing) that tests two hypothesis and the second is an exploratory study (hypothesis-generating). The two types of studies differ in their aims and methodologies, but they are both relevant and were pursued within the same research project.

In the hypothesis-generating exploratory study, we explored a dataset of neonates who were treated with 5 days of intravenous antibiotics to describe the relationships and patterns in the observed noxious-evoked responses, so that we could generate a hypothesis which can be tested in a subsequent prospective confirmatory study. Due to the exploratory nature of this study, we do not use inferential statistics i.e. there is no statistical hypothesis testing and assessments of statistical significance. Instead, we use descriptive statistics, to describe the data for use in future hypothesis construction and testing. The higher mean magnitude of brain activity is a description of the data and an interesting observation that warrants follow-up. If confirmed, it would be consistent with evidence from pre-clinical models (Denk, McMahon, & Tracey, 2014; Zouikr & Karshikoff, 2017). However, we acknowledge that sustained hyperalgesia is not likely to be the only possible interpretation and we have added an alternative interpretation.

We have clarified this important point in the manuscript.

“However, when the magnitude of the noxious-evoked brain activity was compared with postnatal age-matched healthy neonates, with no evidence of infection, who were having a blood test for routine newborn screening (‘Neonatal Antibiotic Control Group’), the magnitude of the brain activity was higher (magnitude of noxious-evoked brain activity: Neonatal Antibiotic Treatment Group: n=12, mean=1.21, SD=1.02; Neonatal Antibiotic Control Group: n=8, mean=0.9, SD=0.59; 34% increase, Figure 3C).”

“One interpretation of this exploratory finding is that hyperalgesia could outlive the acute inflammatory period that caused it, with neonatal inflammation potentially causing long-term changes in pain sensitivity. Our current experimental design does not allow for formal hypothesis testing, but if this trend was to be

confirmed in a future study it would be consistent with observations made by other research groups in pre-clinical models [19,33]. It would also provide a potential underlying biological mechanism for what has been observed in epidemiological studies, which report that chronic pain is more likely in individuals who have experienced adverse events in childhood or adolescence [34]; this includes physical trauma and serious illness, all presumably associated with significant levels of inflammation. However, the effect observed in the exploratory study might be no larger than difference due to random chance. Thus, an alternative interpretation of this observation is that early life infection that has been treated with antibiotics results in no long-term consequences in noxious evoked responses. An appropriately powered confirmatory study is warranted to test the hypothesis derived from this exploratory study.” – pages 6 and 7, lines 171-198.

There were no data presented on clinical scores of pain for this exploratory part of the study and it was not clear if this was done or not.

We do not present data on clinical pain scores as given the results from the hypothesis-testing study, we did not expect to observe trends in the PIPP-R scores in the exploratory study. The calculated PIPP-R scores for the hypothesis testing study are similar between groups (Neonatal Inflammation Group mean = 5.7, Neonatal Control Group mean = 6) and did not justify follow-up analysis.

Information on how the antibiotics were delivered (e.g. intravenously or orally) was also not specified.

We provided a schematic representation of how the antibiotics were delivered in Figure 1. We have also specified the route of administration of antibiotics in the Methods section.

“All neonates in these groups presented with risk factors or clinical indicators of early onset neonatal infection, required a blood test to quantify C-reactive protein (CRP) levels and received the same regimen of intravenous antibiotics (commencing 18-24 hours before the study, see (National Institute for Health and Care Excellence (NICE), 2012) for guidelines that were in place during the study period).– page 7, lines 221-223.

“Neonates receiving a 5-day regimen of intravenous antibiotics due to suspected infection were studied during a blood test required to quantify CRP levels 84 - 96 hours after the first dose of antibiotics; participants with CRP < 10 mg/l at the time of study were included in the Neonatal Antibiotic Treatment Group (n=12) as they previously had raised CRP levels.” – page 8, lines 248-251.

The methods were well described and included clear inclusion/ exclusion criteria. A sample size calculation was included to ensure an adequate sample size for the main hypothesis testing – none was provided for the exploratory study thus we do not know if the sample size was adequate for this exploratory work.

As outlined above, the exploratory study uses descriptive statistics, not inferential statistics. Sample size planning and statistical hypothesis testing are not appropriate for the goals of exploratory work. We have clarified this point in the updated manuscript text.

“Sample size calculations were not performed for the exploratory study as no hypotheses were being tested.” – page 13, lines 436-437.

[Data from] a few infants were lost, due to technical issues; leaving 55 neonates for the analyses – this needs to be further clarified in Table 1.

Table 2 (previously Table 1) presents the demographic data for all the participants included across five different analyses. The Extended Data Figure 2 presents the details of the number of neonates included in the analysis for each individual outcome. Due to technical issues, which are detailed in Extended Data Figure 2, a total of 55 neonates were included in the PIPP-R analysis.

In terms of description of the PIPP-R scores, it is important to state how the 2 contextual variables were calculated and whether these were included in the total PIPP-R scores given the revised scoring

instructions. The PIPP-R scores for full term infants are also calculated out of a total of 18 (not 21). Statistical analyses appears to be appropriate.

We followed the revised scoring instructions to calculate the PIPP-R scores (page 11, line 369). The contextual (postmenstrual age and behavioural state) indicators were scored only when the physiological (heart rate and oxygen saturation) or behavioural (brow bulge, eye squeeze and naso-labial furrow) indicators were scored as > 0 (Stevens BJ et al., 2014). We agree with the reviewer that the maximum PIPP-R score for a term infant is 18; however, we state on page 12 that the maximum PIPP-R score is 21 in reference to the general application of the score. Importantly, this is consistent with the most updated guidelines from the Premature Infant Pain Profile-revised (PIPP-R) e-Learning Module that state that: "The PIPP-R is scored out of 21. Note: The only time the PIPP-R is scored out of 18 is if you cannot see one of the behavioural indicators: naso-labial furrow, brow bulge or eye squeeze." (Stevens, 2021)

Overall: This is an important first study, in looking at the relationship between inflammation and pain in human neonates thus extending our knowledge from animal studies. The methods are well conceived, although there may not be an adequate sample size for the exploratory study. Some details require further clarification as outlined above. The results should be reported without overgeneralizing. The conclusion that an adequately powered prospective study is the next logical step in accurate.

Thank you for your considerate and thoughtful review, and for appreciating the important value of this work.

Reviewer #3:

The Slater group has for a decade or more made substantial contributions to the infant pain literature. They have pioneered EEG, EMG, behavior and brain imaging work in neonates, linking those data closely in some cases to the basic science literature. This is an additional study showing that ongoing infection with antibiotics increases spinal and brain reactivity to noxious and tactile stimuli. The data are solid as are the methods. However, the interpretation of the data is problematic.

We thank the reviewer for their insightful comments and detailed feedback.

First, because most the the preclinical and some of the clinical data on infant infection looks at longer term outcomes that outlast the infection it should be clearly emphasized that the infection is concurrent with the testing. The abstract and intro are accurate but the introduction emphasizes long-term outcomes (which indeed is the bulk of the literature). For example, the first paragraph states outcomes from early infection that occur later in life - asthma, type-1 diabetes, later behavioral, motor and cognitive behavior. The basic science papers cited, especially those of Zouikr, are early injury and later outcomes. Indeed ref. 15, showed that 2 days after LPS there were no changes in a formalin challenge and that it took >2 weeks after LPS challenge to show changes in SDH.

As noted by the reviewer, the short-term effects of neonatal inflammation on pain responses have not been extensively explored in previous clinical and pre-clinical studies as most of the literature investigates the long-term effects of neonatal inflammation. Preclinical studies assessing pain behaviour within the first hours to two days after a neonatal LPS challenge show conflicting results (Hunter, Chai, & Barr, 2015; Lidow, Song, & Ren, 2001; Zouikr et al., 2014). Thus, the mechanisms of the neuroimmune interactions and observed inflammation-induced hyperalgesia are derived from adult human and adult animal studies.

We have clarified these points in the introduction and results.

“There are, for instance, reports that experimental activation of the neonatal immune system in rat pups, via lipopolysaccharide (LPS) administration, causes long-term increased pain sensitivity [12-16]. However, studies investigating how the acute phase of early life inflammation impacts pain sensitivity are scarce. Moreover, given the many known inter-species differences in both immune and nociceptive systems [17], it is of vital importance that we generate data in human neonates, which can be relevantly back-translated into future rodent models.” – page 3, lines 53-59.

“Our findings are thus in line with evidence from adult humans and animals, which shows that neuro-immune interactions increase spinal cord excitability and nociceptive sensitivity [19–22]. However, evidence from neonatal humans and animals is lacking. Pain assessment during the period of neonatal inflammation has not been extensively investigated and studies undertaken in neonatal rat pups report conflicting results [14,23]. Our specific findings in human infants using CRP as a marker of inflammation could be further explored in rodent models to elucidate the potential mechanisms of these neuro-immune interactions.” – pages 4 and 5, lines 106-115.

Also, the fact that the two independent variables are infection and concurrent antibiotic treatment, the title and abstract need to include that. Are there prior data on the effects of antibiotics on pain independent of their anti-bacterial effects. That needs some discussion and is a clinically mandated confound.

We agree with the reviewer that antibiotics could potentially influence the pain responses concomitant with the antibacterial effects. There are, for instance, reports of the role of gut microbiota composition on the regulation of pain processing and other behaviours (Cryan & Dinan, 2012; Mayer, Knight, Mazmanian, Cryan, & Tillisch, 2014). Yan and Kentner investigated the role of microbiota on pain sensitivity and reported that antibiotics reduced allodynia in neonatal LPS treated rats (Yan & Kentner, 2017). While this is a highly valid point which has been appropriately raised by the reviewer, the potential confounding effect of the antibiotic treatment has been accounted for in our experimental design given that all neonates independent of the group allocation received the same regimen of intravenous antibiotics due to their presentation of signs and symptoms of suspected EONI (see Figure 1 and page 3 - lines 63-66). Therefore, in this case we do not think that the presence of antibiotics is substantially impacting the group differences that were observed.

We have clarified this point in the Methods section:

“Participants in the Neonatal Control Group (n=38) and Neonatal Inflammation Group (n=23) were born between 36 and 42 weeks’ gestation and were less than 72 hours old at the time of study (Table 2). All neonates in these groups presented with risk factors or clinical indicators of early onset neonatal infection, required a blood test to quantify C-reactive protein (CRP) levels and received the same regimen of intravenous antibiotics (commencing 18-24 hours before the study, see [18] for guidelines that were in place during the study period).” – page 7, lines 219-225.

Although it is clear that tactile and noxious stimuli produce an exaggerated spinal and brain response, the PIPPS-R data are not different between groups. The argument that “sickness syndrome” had an effect that reduced the validity of the PIPPS-R data certainly may be true. Would that also be expected to alter the EMG? Are there other data showing a discrepancy between PIPPS and biological measurements of pain. Are there data on the PIPPS in other infection studies? To this reviewer, the authors have balanced the biological and behavioral outcomes well but more discussion of other PIPPS data might be clarifying.

Given that the neural mechanisms underpinning limb withdrawal and facial expression are different, it is possible that sickness syndrome affects these motor circuits differently, such that limb withdrawal circuits are sensitised by inflammation, but facial expression circuits are not. However, it is also possible that inflammation affects limb withdrawal and facial expressions to the same degree, but we were not able to detect an effect due to differences between assessment methods. The limb withdrawal is assessed with EMG, which is objective and sensitive to differences that would not be identified visually. The facial

expression is assessed by visual inspection, which is subjective and potentially noisier due to inter- and intra-rater variability. Thus, it is possible that we are underpowered to observe an effect of inflammation on clinical pain scores given the limitations of the PIPP-R assessment method. Further research is needed to explore the impact of inflammation on pain behaviour in neonates.

We have edited page 5 to provide a more nuanced discussion of this point.

“While it is possible that we are underpowered to observe an effect on clinical pain scores (given the subjective nature of PIPP assessments in contrast to objective EMG/EEG assessments), this observation is highly consistent with clinical observations in adults and neonates, and data from animal studies, which demonstrate that inflammation suppresses behavioural activity, causing listlessness and lethargy [24,25].” - page 5, lines 122-126.

“Further research is warranted to establish how the presence of neonatal inflammation impacts clinical pain scores and the degree to which subjective observational assessment of infant pain-related behaviours might influence pain scoring.” - page 5, lines 134-137.

We are not aware of previous literature using PIPP scores or other pain scales during infection in infants. However, the observation that the EMG activity conversely increases during inflammation is interesting and supported by current literature. Studies suggest that muscle reflexes are exaggerated in the presence of inflammation, and that increased excitability within dorsal horn networks could result in large EMG responses. This observation is also consistent with clinical presentations which suggest that increased irritability and restlessness are observed during inflammatory conditions (see page 6, lines 158-162).

The hypothesis explorative study, 5 days after treatment or not with antibiotics, shows NO difference despite the statement that evoked brain activity was higher, albeit not significantly. The authors conclude "the hyperalgesia could outlive the acute inflammatory period". This statement is problematic and implies to this reader, a trend towards significance. But statistical analysis given the mean,SD and N of each group showed a p-value of ~0.45, so clearly the two groups did NOT differ. Indeed the proper argument given these preliminary data is more reassuring - early life treated infection (or antibiotic treatment) has no long term consequence for noxious evoked brain activity.

We agree with the reviewer and acknowledge that the exploratory study results were not described clearly. Given that we are not specifically testing any hypothesis in the exploratory study, statistical hypothesis testing is not appropriate. Thus, concrete interpretation of the results as supporting the null hypothesis (no long-term consequences) or at odds with the null hypothesis (potential for long-term consequences) is not warranted. Instead, our presentation of descriptive (non-inferential) statistics and the subsequent discussion of these data descriptions (and how they might motivate future studies) was the intended goal. It was not our intention to conclude that hyperalgesia does indeed outlive the acute inflammatory period; rather that this is an interesting hypothesis that the descriptive statistics could be suggesting, and this hypothesis is worth assessing in a future confirmatory study. We agree however that sustained hyperalgesia is not likely to be the only possible interpretation and we have added the alternative interpretation which has been suggested by the reviewer.

“However, when the magnitude of the noxious-evoked brain activity was compared with postnatal age-matched healthy neonates, with no evidence of infection, who were having a blood test for routine newborn screening (‘Neonatal Antibiotic Control Group’), the magnitude of the brain activity was higher (magnitude of noxious-evoked brain activity: Neonatal Antibiotic Treatment Group: n=12, mean=1.21, SD=1.02; Neonatal Antibiotic Control Group: n=8, mean=0.9, SD=0.59; 34% increase, Figure 3C).”

“One interpretation of this exploratory finding is that hyperalgesia could outlive the acute inflammatory period that caused it, with neonatal inflammation potentially causing long-term changes in pain sensitivity. Our current experimental design does not allow for formal hypothesis testing, but if this trend was to be confirmed in a future study it would be consistent with observations made by other research groups in pre-clinical models (Denk et al., 2014; Zouikr & Karshikoff, 2017). It would also provide a potential underlying

biological mechanism for what has been observed in epidemiological studies, which report that chronic pain is more likely in individuals who have experienced adverse events in childhood or adolescence (Jones, Power, & Macfarlane, 2009); this includes physical trauma and serious illness, all presumably associated with significant levels of inflammation. However, the effect observed in the exploratory study might be no larger than difference due to random chance. Thus, an alternative interpretation of this observation is that early life infection that has been treated with antibiotics results in no long-term consequences in noxious evoked responses. An appropriately powered confirmatory study is warranted to test the hypothesis derived from this exploratory study.” – pages 6 and 7, lines 171-200.

Minor point - a bit of discussion on the possible effects of comforting the infant with swaddling and/or non-nutritive sucking. Perhaps the comforting affects the PIPP-R data?

It is possible that comforting measures have an effect on the clinical pain scores; however, infants in both groups received the same comforting measures as stated in the Methods section (page 9, lines 272-274) “Neonates were only studied if a heel lance was necessary as part of their clinical care to assess their CRP levels or for routine screening. As standard practice, the comfort of the neonates was prioritised during the study with the use of swaddling and/or non-nutritive sucking.”. Therefore, this is unlikely to be a confound factor in our analysis.

Thus, my major critique is not of the data or methods but the slant put on those data in their interpretation. One could easily argue that infection+antibiotics, although concurrently altering EMG and EEG, has no behavioral consequence and that the altered EEG is transient at best. This is a different take on the data than that of the authors, but one that I think is equally valid.

Thank you for your thoughtful review of this manuscript and for proposing valid and thoughtful alternative explanations that could explain the reported observations.

References

- Altman, D. G., & Royston, P. (2006). The cost of dichotomising continuous variables. *British Medical Journal*, 332(7549), 1080. <https://doi.org/10.1136/bmj.332.7549.1080>
- Andrews, K., & Fitzgerald, M. (1994). The cutaneous withdrawal reflex in human neonates: sensitization, receptive fields, and the effects of contralateral stimulation. *Pain*, 56(1), 95–101. [https://doi.org/10.1016/0304-3959\(94\)90154-6](https://doi.org/10.1016/0304-3959(94)90154-6)
- Andrews, K., & Fitzgerald, M. (1999). Cutaneous flexion reflex in human neonates: A quantitative study of threshold and stimulus-response characteristics after single and repeated stimuli. *Developmental Medicine and Child Neurology*, 41(10), 696–703. <https://doi.org/10.1017/S0012162299001425>
- Barr, G. A., & Hunter, D. A. (2014). Interactions between glia, the immune system and pain processes during early development. *Developmental Psychobiology*, 56(8), 1698–1710. <https://doi.org/10.1002/dev.21229>
- Cobo, M. M., Hartley, C., Gursul, D., Andritsou, F., van der Vaart, M., Schmidt Mellado, G., ... Slater, R. (2021). Quantifying individual noxious-evoked baseline sensitivity to optimise analgesic trials in neonates. *ELife*, 10, 1–24. <https://doi.org/10.7554/elife.65266>
- Cornelissen, L., Fabrizi, L., Patten, D., Worley, A., Meek, J., Boyd, S., ... Fitzgerald, M. (2013). Postnatal temporal, spatial and modality tuning of nociceptive cutaneous flexion reflexes in human infants. *PLoS ONE*, 8(10), e76470. <https://doi.org/10.1371/journal.pone.0076470>
- Cryan, J. F., & Dinan, T. G. (2012). Mind-altering microorganisms: The impact of the gut microbiota on brain and behaviour. *Nature Reviews Neuroscience*, 13(10), 701–712. <https://doi.org/10.1038/nrn3346>
- Denk, F., McMahon, S. B., & Tracey, I. (2014). Pain vulnerability: a neurobiological perspective. *Nature Neuroscience*, 17(2), 192–200. <https://doi.org/10.1038/nn.3628>
- Goksan, S., Hartley, C., Emery, F., Cockrill, N., Poorun, R., Moultrie, F., ... Slater, R. (2015). fMRI reveals neural activity overlap between adult and infant pain. *ELife*, 4, e06356.

<https://doi.org/10.7554/eLife.06356>

- Gursul, D., Goksan, S., Hartley, C., Mellado, G. S., Moultrie, F., Hoskin, A., ... Slater, R. (2018). Stroking modulates noxious-evoked brain activity in human infants. *Current Biology*, 28(24), R1380–R1381. <https://doi.org/10.1016/j.cub.2018.11.014>
- Hartley, C., Duff, E. P., Green, G., Mellado, G. S., Worley, A., Rogers, R., & Slater, R. (2017). Nociceptive brain activity as a measure of analgesic efficacy in infants. *Science Translational Medicine*, 9(388). <https://doi.org/10.1126/scitranslmed.aah6122>
- Hartley, C., Goksan, S., Poorun, R., Brotherhood, K., Mellado, G. S., Moultrie, F., ... Slater, R. (2015). The relationship between nociceptive brain activity, spinal reflex withdrawal and behaviour in newborn infants. *Scientific Reports*, 5(12519). <https://doi.org/10.1038/srep12519>
- Hartley, C., Moultrie, F., Gursul, D., Hoskin, A., Adams, E., Rogers, R., & Slater, R. (2016). Changing balance of spinal cord excitability and nociceptive brain activity in early human development. *Current Biology*, 26(15), 1998–2002. <https://doi.org/10.1016/j.cub.2016.05.054>
- Hartley, C., Moultrie, F., Hoskin, A., Green, G., Monk, V., Bell, J., ... Slater, R. (2018). Analgesic efficacy and safety of morphine in the Procedural Pain in Premature Infants (Poppi) study: randomised placebo-controlled trial. *The Lancet*, 392(10164), 2595–2605. [https://doi.org/10.1016/S0140-6736\(18\)31813-0](https://doi.org/10.1016/S0140-6736(18)31813-0)
- Hunter, D., Chai, C., & Barr, G. A. (2015). Effects of COX inhibition and LPS on formalin induced pain in the infant rat. *Developmental Neurobiology*, 75(10), 1068–1079. <https://doi.org/10.1002/dneu.22230>
- Jones, G. T., Power, C., & Macfarlane, G. J. (2009). Adverse events in childhood and chronic widespread pain in adult life: Results from the 1958 British Birth Cohort Study. *Pain*, 143(1–2), 92–96. <https://doi.org/10.1016/j.pain.2009.02.003>
- Kasser, S., Hartley, C., Rickenbacher, H., Klarer, N., Depoorter, A., Datta, A. N., ... Wellmann, S. (2019). Birth experience in newborn infants is associated with changes in nociceptive sensitivity. *Scientific Reports*, 9(1), 4117. <https://doi.org/10.1038/s41598-019-40650-2>
- Kühne, B., Stützer, H., Roth, B., & Welzing, L. (2012). The flexion withdrawal reflex reveals an increasing threshold during the first year of life and is influenced by the infant's state of consciousness. *Klinische Padiatrie*, 224(5), 291–295. <https://doi.org/10.1055/s-0031-1301328>
- Lidow, M. S., Song, Z. M., & Ren, K. (2001). Long-term effects of short-lasting early local inflammatory insult. *NeuroReport*, 12(2), 399–403. <https://doi.org/10.1097/00001756-200102120-00042>
- Mayer, E. A., Knight, R., Mazmanian, S. K., Cryan, J. F., & Tillisch, K. (2014). Gut microbes and the brain: Paradigm shift in neuroscience. *Journal of Neuroscience*, 34(46), 15490–15496. <https://doi.org/10.1523/JNEUROSCI.3299-14.2014>
- National Institute for Health and Care Excellence (NICE). (2012). *Antibiotics for early-onset neonatal infection : antibiotics for the prevention and treatment of early-onset neonatal infection*. (August), NICE Guideline No 149.
- Slater, R., Cantarella, A., Gallella, S., Worley, A., Boyd, S., Meek, J., & Fitzgerald, M. (2006). Cortical pain responses in human infants. *The Journal of Neuroscience*, 26(14), 3662–3666. <https://doi.org/10.1523/jneurosci.0348-06.2006>
- Slater, R., Worley, A., Fabrizi, L., Roberts, S., Meek, J., Boyd, S., & Fitzgerald, M. (2010). Evoked potentials generated by noxious stimulation in the human infant brain. *European Journal of Pain*, 14(3), 321–326. <https://doi.org/10.1016/j.ejpain.2009.05.005>
- Sproston, N. R., & Ashworth, J. J. (2018). Role of C-reactive protein at sites of inflammation and infection. *Frontiers in Immunology*, 9(APR), 1–11. <https://doi.org/10.3389/fimmu.2018.00754>
- Stevens, B. (2021). Premature Infant Pain Profile-revised (PIPP-R) e-Learning Module. Retrieved from The Hospital for Sick Children ("SickKids") - Academy Online website: <https://lab.research.sickkids.ca/stevens/pipp-r-module/>
- Stevens BJ, Gibbins S, Yamada J, Dionne K, Lee G, Johnston C, & Al, E. (2014). The Premature Infant Pain Profile-Revised (PIPP-R): Initial Validation and Feasibility. *Clin J Pain*, 30(3), 238–243. <https://doi.org/10.1097/AJP.0b013e3182906aed>
- van der Vaart, M., Duff, E., Raafat, N., Rogers, R., Hartley, C., & Slater, R. (2019). Multimodal pain assessment improves discrimination between noxious and non-noxious stimuli in infants. *Paediatric and Neonatal Pain*, 1(1), 21–30. <https://doi.org/10.1002/pne2.12007>
- Yan, S., & Kentner, A. C. (2017). Mechanical allodynia corresponds to Oprm1 downregulation within the descending pain network of male and female rats exposed to neonatal immune challenge. *Brain*,

Behavior, and Immunity, 63, 148–159. <https://doi.org/10.1016/j.bbi.2016.10.007>

Zouikr, I., & Karshikoff, B. (2017). Lifetime modulation of the pain system via neuroimmune and neuroendocrine interactions. *Frontiers in Immunology*, 8(MAR).
<https://doi.org/10.3389/fimmu.2017.00276>

Zouikr, I., Tadros, M. A., Barouei, J., Beagley, K. W., Clifton, V. L., Callister, R. J., & Hodgson, D. M. (2014). Altered nociceptive, endocrine, and dorsal horn neuron responses in rats following a neonatal immune challenge. *Psychoneuroendocrinology*, 41, 1–12.
<https://doi.org/10.1016/j.psyneuen.2013.11.016>

Reviewers' Comments:

Reviewer #1:

Remarks to the Author:

No further comments.

Reviewer #2:

Remarks to the Author:

The authors have addressed most of the issues raised by the reviewers in a reasonable way. I still would like to see a more detailed response for the following:

1. the authors state that they have dichotomized the CRP data, which is continuous in nature. The dichotomization, although consistent with clinical decision making, is still somewhat arbitrary (e.g. if the cut off point had been different than $>10\text{mg/l}$; then the results could have been different). It would be interesting to know if there were any significant results, had the continuous CRP data been used. The authors need to address this point and provide better rationale for using dichotomized rather than continuous data; in light of the limited sample size, modest p values and power.
2. the issue of how the presence of neonatal inflammation impacts clinical pain scores (PIPP-R) is still not clearly articulated. I believe that the issue may be more fundamental than the adequacy of the pain score itself - rather what is the influence of inflammation on different indicators of pain including behavioral (facial expression indicators, limb movement, cry), physiologic indicators, neurologic indicators). Although this may not be the main focus of this study, since the authors emphasize that this work was exploratory, it would be ideal to complete this work to determine how inflammation affects each of these indicators.
3. when comparing the magnitude of noxious-evoked brain activity in infants with inflammation and the historical age-matched healthy controls, it is important to emphasize that there was no statistical difference between the two groups, as the authors have clearly stated that this is an observational study with no statistical comparisons.

Reviewer #3:

Remarks to the Author:

The authors have responded in detail to my comments on the initial review. I would emphasize some of the conclusions from the data differently (mainly the exploratory study) because although they are not hypothesis testing, they indeed do have an implicit hypothesis. But I think that this is a reasonable disagreement and they do provide sufficient data that the reader can draw their own conclusions.

A suggestion, not a requisite, for this reviewer: Given the scatter of the data from the primary study, an estimation plot would help the reader understand the magnitude of the statistical differences. See "Moving beyond P values: data analysis with estimation graphics" by Jose Ho, Tayfun Tumkaya, Sameer Aryal, Hyungwon Choi & Adam Claridge-Chang *Nature Methods* volume 16, pages 565–566 (2019). (sorry for not suggesting this in the original critique). Also, a fairly recent basis science study looked at long- and short-term effects of E-coli infection with and without injury on various measures of pain and pain affect ("Local injury and systemic infection in infants alter later nociception and pain affect during early life and adulthood", C.I. Gomes and G.A. Barr, *Brain, Behavior and Immunity- Health*, 9 (10075), 2020) that would be consistent with the authors exploratory study.

Point-by-point response to the reviewers' comments

Reviewer Comments

Reviewer #1 (Remarks to the Author):

No further comments.

Thank you for your thoughtful review and help in improving the clarity of the manuscript and our data interpretation.

Reviewer #2 (Remarks to the Author):

The authors have addressed most of the issues raised by the reviewers in a reasonable way. I still would like to see a more detailed response for the following:

1. the authors state that they have dichotomized the CRP data, which is continuous in nature. The dichotomization, although consistent with clinical decision making, is still somewhat arbitrary (e.g. if the cut off point had been different than $>10\text{mg/l}$; then the results could have been different). It would be interesting to know if there were any significant results, had the continuous CRP data been used. The authors need to address this point and provide better rationale for using dichotomized rather than continuous data; in light of the limited sample size, modest p values and power.

We agree that the clinical cut off that drives the decision to administer antibiotics is relatively arbitrary and that the results of the analysis are directly dependent on this value. It is for this reason we performed and report a full statistical analysis of the continuous data (Figure 2b, 2d). We consider that this point is fully addressed in the manuscript (page 4, lines 92-96):

“Due to the continuous nature of the CRP level variable, we assessed correlations between CRP level and EMG and EEG responses. We show that the spinal cord mediated reflex withdrawal activity and noxious-evoked brain activity significantly correlated with the level of CRP in the blood sample (reflex withdrawal activity: $n=57$, Pearson correlation $r=0.28$, $p=0.028$, noxious-evoked brain activity: $n=51$, $r=0.31$, $p=0.023$, Figure 2).”

2. the issue of how the presence of neonatal inflammation impacts clinical pain scores (PIPP-R) is still not clearly articulated. I believe that the issue may be more fundamental than the adequacy of the pain score itself - rather what is the influence of inflammation on different indicators of pain including behavioral (facial expression indicators, limb movement, cry), physiologic indicators, neurologic indicators). Although this may not be the main focus of this study, since the authors emphasize that this work was exploratory, it would be ideal to complete this work to determine how inflammation affects each of these indicators.

Thank you for raising this important point. We agree that the impact of inflammation on the individual components of the PIPP-R score is fundamentally important. As requested, we have therefore taken each noxious-evoked response which we have measured: (i) noxious-evoked brain activity, (ii) reflex activity, (iii) brow bulge duration, (iv) nasolabial furrow duration, (v) eye squeeze duration, (vi) oxygen saturation change; and (vii) heart rate change and provide the t-statistics, Pearson correlation coefficients and p-values for both the dichotomised and continuous data. This data is included in a new Supplementary Table 1 (below). The text in the manuscript has not been edited as this supplementary data is in line with the main results which are already described and discussed in the main manuscript text. Thank you for this helpful suggestion.

“Supplementary Table 1. Relationship between CRP levels (dichotomised and continuous) and noxious-evoked responses.

	EMG	EEG	Brow bulge	Eye squeeze	Nasolabial furrow	Heart rate change	Oxygen saturation change
T-test	1.74 (0.048)	2.4 (0.011)	0.53 (0.29)	0.53 (0.29)	0.72 (0.23)	-0.74 (0.77)	0.62 (0.30)
Correlation	0.28 (0.028)	0.31 (0.023)	0.05 (0.34)	0.03 (0.39)	0.07 (0.31)	-0.11 (0.79)	0.14 (0.13)

Brow bulge, eye squeeze, nasolabial furrow, heart rate change and oxygen saturation change correspond to the individual components of the PIPP-R scores (hypothesis-testing study).

T-test: t-statistics and one-sided uncorrected p-values (parentheses) are presented for each metric.

Correlation: Pearson correlation coefficients and one-sided uncorrected p-values (parentheses) are presented for each metric.”

3. when comparing the magnitude of noxious-evoked brain activity in infants with inflammation and the historical age-matched healthy controls, it is important to emphasize that there was no statistical difference between the two groups, as the authors have clearly stated that this is an observational study with no statistical comparisons.

As implied by the reviewer we have ‘clearly stated that this is an observational study with no statistical comparisons’ therefore it is not possible to ‘emphasise that there was no statistical difference between the two groups’ as this would implicitly require the use of inappropriate statistical tests. Nevertheless, we appreciate the exploratory nature of this study, which will facilitate further hypothesis testing studies, and have emphasised this throughout the manuscript. An additional comment has been added to the manuscript (page 6; lines 173-175).

“As the neonates in each group were only matched in terms of age in this exploratory study, other factors such as prior pain experience or severity of illness were not necessarily comparable between groups, which limits the interpretation of this finding.”

Reviewer #3 (Remarks to the Author):

The authors have responded in detail to my comments on the initial review. I would emphasize some of the conclusions from the data differently (mainly the exploratory study) because although they

are not hypothesis testing, they indeed do have an implicit hypothesis. But I think that this is a reasonable disagreement and they do provide sufficient data that the reader can draw their own conclusions.

Thank you for your thoughtful reviewer comments, which have substantially improved the clarity of the manuscript and framing of the results.

A suggestion, not a requisite, for this reviewer: Given the scatter of the data from the primary study, an estimation plot would help the reader understand the magnitude of the statistical differences. See "Moving beyond P values: data analysis with estimation graphics" by Joes Ho, Tayfun Tumkaya, Sameer Aryal, Hyungwon Choi & Adam Claridge-Chang Nature Methods volume 16, pages 565–566 (2019). (sorry for not suggesting this in the original critique). Also, a fairly recent basis science study looked at long- and short-term effects of E-coli infection with and without injury on various measures of pain and pain affect ("Local injury and systemic infection in infants alter later nociception and pain affect during early life and adulthood", C.I. Gomes and G.A. Barr, Brain, Behavior and Immunity- Health, 9 (10075), 2020) that would be consistent with the authors exploratory study.

Thank you for the suggestion of using estimation graphics, this is a great idea. However, using the estimation plots is not ideal for this study design as our hypotheses are directional and the t-tests are one-sided. Instead of presenting the proposed figures, we have revised the main text to include the estimation statistics (i.e. the difference in sample means and the 95% confidence interval).

"Newborn infants in the Neonatal Inflammation Group had significantly greater spinal cord mediated reflex withdrawal EMG activity and noxious-evoked EEG brain activity in response to heel lancing (within 24 hours from presentation of risk factors) compared to neonates who do not have raised inflammatory markers (magnitude of noxious-evoked reflex withdrawal activity: Neonatal Inflammation Group: n=21, mean=42.1, SO=30.9; Neonatal Control Group: n=36, mean=30.2, SO=20.6; A=11.9, 95%CI=[1.32, x], t-test t=1.74, p=0.048; magnitude of noxious-evoked brain activity: Neonatal Inflammation Group: n=19, mean=1.04, SO=0.68; Neonatal Control Group: n=32, mean=0.64, SO=0.52; A=0.41, 95%CI=[0.13, x], t=2.4, p=0.022, p-values corrected for multiple comparisons using Holm's method, Table 1, Figure 2)." Page 4, lines 84 – 92

"As hypothesised, neonatal inflammation evoked increased reflex withdrawal activity to tactile stimulation (magnitude of reflex withdrawal: Neonatal Inflammation Group: n=21, mean=17.9, SO=12.2; Neonatal Control Group: n=36, mean=11.8, SO=12; A=6.15, 95%CI=[0.92, x], t=1.85, p=0.037, Figure 3A) and significantly greater brain activity following tactile stimulation (magnitude of tactile-evoked brain activity: Inflammation Group, n=19, mean=0.1, SO=0.09; Control Group, n=29, mean=0.05, SO=0.05; A=0.05, 95%CI=[0.02, x], t=2.66, p=0.007, Figure 3B)." Pages 5 -6, lines 146152.

In addition, we agree that Gomes and Barr (2020) is a highly relevant publication and their results are consistent with the exploratory study presented in this manuscript. It was an oversight not to include it in the original manuscript. This paper is now appropriately referenced (page 6, line 176).

"Our current experimental design does not allow for formal hypothesis testing, but if this trend was to be confirmed in a future study, it would be consistent with observations made by other research groups in pre-clinical models 19,32,33."

Reviewers' Comments:

Reviewer #2:

Remarks to the Author:

Thank you to the authors for responding to my comments and suggestions. I have no further comments for the authors.

Reviewer #3:

Remarks to the Author:

The authors have responded to the last round of comments and I have no further criticisms of this work.